# Circulating transforming growth factor-β1 facilitates remyelination in the adult central nervous system

Machika Hamaguchi[1], Rieko Muramatsu[1,2,3]*, Harutoshi Fujimura[4], Hideki Mochizuki[5], Hirotoshi Kataoka[6], Toshihide Yamashita[1,3,6,7]

[1]Department of Molecular Neuroscience, Graduate School of Medicine, Osaka University, Suita, Japan; [2]Department of Molecular Pharmacology, National Institute of Neuroscience, National Center of Neurology and Psychiatry, Kodaira, Japan; [3]WPI Immunology Frontier Research Center, Osaka University, Suita, Japan; [4]Toneyama National Hospital, Toyonaka, Japan; [5]Department of Neurology, Graduate School of Medicine, Osaka University, Suita, Japan; [6]Department of Neuro-Medical Science, Graduate School of Medicine, Osaka University, Suita, Japan; [7]Graduate School of Frontier Biosciences, Osaka University, Suita, Japan

**Abstract** Oligodendrocyte maturation is necessary for functional regeneration in the CNS; however, the mechanisms by which the systemic environment regulates oligodendrocyte maturation is unclear. We found that Transforming growth factor (TGF)-β1, which is present in higher levels in the systemic environment, promotes oligodendrocyte maturation. Oligodendrocyte maturation was enhanced by adult mouse serum treatment via TGF-β type I receptor. Decrease in circulating TGF-β1 level prevented remyelination in the spinal cord after toxin-induced demyelination. TGF-β1 administration promoted remyelination and restored neurological function in a multiple sclerosis animal model. Furthermore, TGF-β1 treatment stimulated human oligodendrocyte maturation. These data provide the therapeutic possibility of TGF-β for demyelinating diseases.
DOI: https://doi.org/10.7554/eLife.41869.001

*For correspondence:
muramatsu@ncnp.go.jp

**Competing interests:** The authors declare that no competing interests exist.

## Introduction

Demyelination, a hallmark of many central nervous system (CNS) diseases, is a main cause of neurological dysfunction. Improved remyelination is of potential interest in driving therapeutic manipulation, but drugs which mediate a fundamental cure for CNS demyelination diseases are not clinically available and are greatly needed. Remyelination is mediated by oligodendrocyte precursor cells (OPCs), which are distributed throughout the adult mammalian CNS (*Dawson et al., 2003*). Successful remyelination requires OPC proliferation, migration, differentiation to oligodendrocytes, and finally, maturation into myelinating oligodendrocytes (*Franklin and Ffrench-Constant, 2017*). The last of these processes is essential for the structural repair and functional integrity of neuronal networks in the CNS.

Remyelination occurs as a spontaneous and efficient process in experimental models and many clinical conditions (*Franklin, 2002*). Because remyelination is controlled by environmental signals (*Clemente et al., 2013*), OPCs and oligodendrocytes around demyelinated lesions are thought to receive the signals which promote maturation. With respect to the molecular mechanism, it is believed that oligodendrocyte maturation is controlled by the CNS-cell-derived factors, such as axonal F3-mediated Notch signaling (*Hu et al., 2003*), and astrocyte-derived leukemia inhibitory factor

(*Fischer et al., 2014*). In contrast, demyelination often occurs simultaneously with vascular damage (*Ruckh et al., 2012*) and vascular damage leads to leakage of circulating factors into the CNS, suggesting that oligodendrocytes around the demyelinating lesion are exposed to circulating factors. In support of this idea, circulating FGF21 leaks into the CNS after injury and promotes OPC proliferation (*Kuroda et al., 2017*), a first step in remyelination. Although previous findings established the concept that successful remyelination requires oligodendrocyte maturation, the mechanism by which the circulating factors regulates oligodendrocyte maturation remains unknown.

TGF-βs are members of a superfamily of multifunctional cytokines with key functions in development, patterning, and immune responses. In CNS pathology, TGF-βs have implications in inflammatory responses through the activation of microglia in a number of animal models of diseases including stroke (*Iadecola and Anrather, 2011*), Alzheimer disease (AD) (*Wyss-Coray, 2006*), and multiple sclerosis (MS) (*Benveniste, 1997*). These observations are in line with the concept that TGF-βs are produced by the CNS and effect physiological reactions in neighboring cells, resulting in the progress of disease severity. However, in experimental autoimmune encephalomyelitis (EAE), systemic administration of TGF-β1 prevents disease severity (*Kuruvilla et al., 1991*). Moreover, the levels of TGF-β1 in serum obtained from the patients with amnestic mild cognitive impairment is low when compared with control patients, and this change is positively correlated with cognitive performance (*Huang et al., 2013*). Because TGF-β is known to control the timing of remyelination (*Palazuelos et al., 2014*), we hypothesized that circulating TGF-β1 may have a protective role in the CNS and may specifically promote remyelination.

In this study, we showed that TGF-β1, which is largely expressed in the peripheral environments, promoted oligodendrocyte maturation in mice spinal cord which were subjected to toxin-induced demyelination. Decrease in circulating TGF-β1 levels prevented spontaneous remyelination. Systemic administration of TGF-β1 promoted remyelination and decreased disease severity in EAE. TGF-β1 treatment promoted expression of myelin-associated gene in human oligodendrocyte in culture. These data suggest that the administration of systemic TGF-β1 may provide a therapeutic avenue for demyelinating diseases.

## Results

### TGF-β1 in adult mouse serum facilities oligodendrocyte maturation in vitro

We first investigated the possibility that circulating factors promote oligodendrocyte maturation. To test this, we performed an in vitro analysis to determine if treatment with adult mouse serum enhances the expression of myelin basic protein (MBP) in Olig2-positive oligodendrocyte lineages cells using high-content imaging analysis (HCA; *Figure 1A*). Adult mouse serum treatment promoted MBP expression in oligodendrocyte-lineage cells (*Figure 1B,C*), indicating that adult serum contains factors involved in the promotion of oligodendrocyte maturation. The MBP expression activity was not abolished by the pre-treatment of serum with DNase, RNase, and heat shock (*Figure 1—figure supplement 1A–D*), suggesting that the factor(s) in the adult mouse serum which may be involved in oligodendrocyte maturation may comprise peptides and have the property of being heat-resistant.

To investigate the molecular mechanism of adult mouse serum-mediated oligodendrocyte maturation, we conducted pharmacological screening (as described in Materials and methods). We first selected drugs that did not inhibit cell survival by counting the number of Olig2-positive cells. Of 160 drugs, 107 drugs did not decrease of the abundance of Olig2-positive cells (0.5 times lower than control) in the culture after each treatment. Next, we excluded the drugs that decreased MBP expression (0.5 times lower than control) in the cells by drugs treatment with or without serum, and identified 17 drugs that caused downregulation of MBP area (two times lower than control) relative to serum treatment (*Figure 1—source data 1*). Because we hypothesized that a circulating molecule drives oligodendrocyte maturation directly, we focused on the TGF-β receptor type I inhibitor (TGF-βRI), the only drug we tested that specifically targets a receptor-type protein. To investigate the role of TGF-βRI in serum-mediated oligodendrocyte maturation, we used LY-364947, a TGF-β type I receptor (TGF-βRI) kinase inhibitor and found that it blocked serum-promoted oligodendrocyte maturation (*Figure 1D,E*). We also confirmed that transfection of oligodendrocytes with TGF-βRI siRNA diminished serum-mediated maturation (*Figure 1F,G*, *Figure 1—figure supplement 1E*). Since TGF-

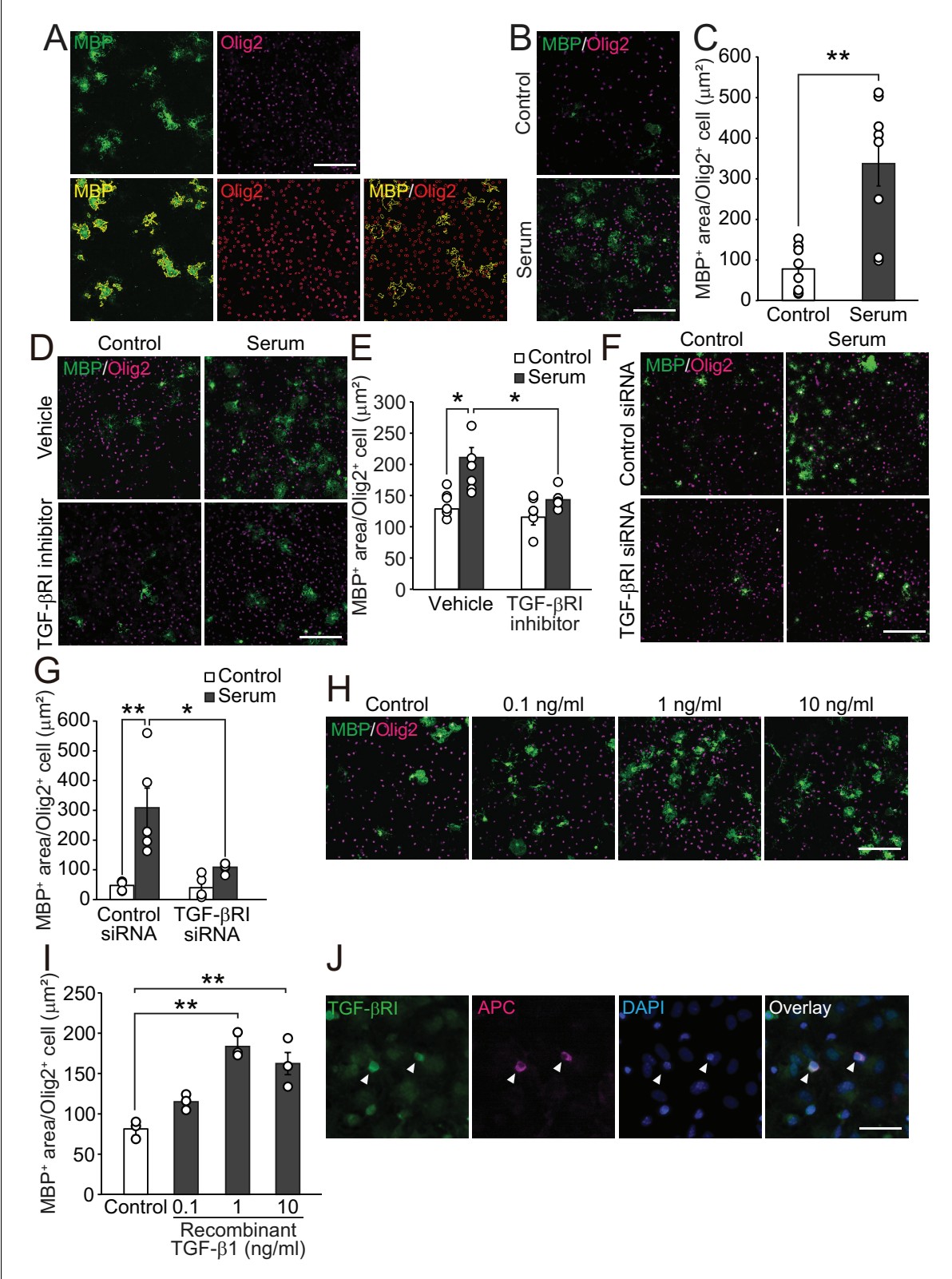

**Figure 1.** TGF-β1 drives oligodendrocyte maturation. (**A**) (Upper panels) Representative images of the oligodendrocyte cultures. Cells were stained with MBP (green) and Olig2 (magenta). (Lower panels) Yellow outlines were obtained from HCA to measure the MBP-positive area in Olig2-positive cells. (**B**) Representative images of the culture stained with MBP and Olig2. Cells were cultured 5 days after the adult mice serum treatment. (**C**) Quantification of MBP-positive area in Olig2-positive cells indicated in B (n = 8 for each), p=0.0009. (**D**) Representative images of culture stained with MBP and Olig2.
*Figure 1 continued on next page*

*Figure 1 continued*

Cells were treated with 1 μM of LY364847 (a TGF-βRI inhibitor), and then with adult mice serum. (E) Quantification of MBP-positive area in Olig2-positive cells indicated in D (n = 6 for each), p=0.0111, 0.0195 (left to right). (F) Representative images of oligodendrocyte stained with MBP and Olig2. Cells were transfected with TGF-β RI siRNA and treated with serum. (G) Quantification of MBP-positive area in Olig2-positive cells indicated in F (n = 5 for each), p=0.0011, 0.0104 (left to right). (H) Representative images of culture stained with MBP and Olig2. Cells were cultured with the indicated concentration of recombinant TGF-β1. (I) Quantification of MBP-positive area in Olig2-positive cells indicated in H (n = 3 for each), p=0.0007, 0.003 (left to right). (J) Representative image of TGF-βRI expression in the culture. Arrowheads indicate cells co-labeled with TGF-βRI (green) and APC (magenta). **p<0.01, *p<0.05, Student's *t*-test or ANOVA with Tukey's post-hoc test. Error bars represent SEM. Scale bars represent 20 μm for J, 100 μm for others.

DOI: https://doi.org/10.7554/eLife.41869.002

The following source data and figure supplement are available for figure 1:

**Source data 1.** Pharmacological impact for serum-mediated oligodendrocyte maturation.
DOI: https://doi.org/10.7554/eLife.41869.004

**Figure supplement 1.** A heat-resistant protein in circulation promotes oligodendrocyte maturation.
DOI: https://doi.org/10.7554/eLife.41869.003

β is known as heat stable protein (*Saita et al., 1994*), we wondered whether TGF-β in serum binds oligodendrocytes directly, resulting in the promotion of oligodendrocyte maturation. We observed that oligodendrocyte treated with recombinant mouse TGF-β1 showed increased maturation (*Figure 1H,I*). Immunocytochemical analysis revealed that adenomatous polyposis coli (APC)-labeled cells in the oligodendrocyte culture expressed TGF-βRI (*Figure 1J*). Western blot analysis revealed that treatment with TGF-β enhanced Smad2 phosphorylation in the oligodendrocyte culture (*Figure 1—figure supplement 1F,G*). These data suggest that TGF-β in adult mouse enhances oligodendrocyte maturation directly.

## Circulating TGF-β1 contributes to remyelination in the CNS

To determine the effect of TGF-β1 in vivo, we investigated the expression pattern of TGF-β1 in adult mice. Quantitative protein analysis revealed that there was appreciable TGF-β1 expression in the spleen (*Figure 2A*), which is correlated with the high concentration of circulating TGF-β1 compared with concentrations in cerebrospinal fluid (CSF) (Serum, 143361.3 ± 18715.5 pg/ml; CSF, 33.4 ± 10.1 pg/ml, *Figure 2B*). To investigate the contribution of peripheral TGF-β1 in CNS remyelination, we used a lysophosphatidylcholine (LPC)-induced demyelination model which accompanied by a vascular barrier disruption around the lesion site (*Muramatsu et al., 2015*). In this model, we detected high levels of TGF-β1 around the LPC lesion in the CNS (*Figure 2C*) without any change in levels of circulating TGF-β1 after LPC injection (*Figure 2D*). These data support the possibility that circulating TGF-β1 accumulates in CNS lesions.

We next investigated whether influx of circulating TGF-β1 into the CNS contributes to the process of remyelination. To analyze the role of circulating TGF-β1 on CNS remyelination, we depleted platelets by injecting anti-CD41 monoclonal antibodies (mAbs) (*Nocito et al., 2007*), as circulating TGF-β1 is transported by platelets (*Grainger et al., 2000*) *Figure 3A*). Platelet depletion reduced the TGF-β1 level in the spleen (*Figure 3—figure supplement 1A*) and the fluorescence intensity of TGF-β1 at the LPC lesion (*Figure 3—figure supplement 1B,C*). We then asked whether a decrease in circulating TGF-β1 level prevents remyelination after LPC injection. Mice, which had registered demyelination immediately after LPC injection, exhibited significant remyelination within 2 weeks post-injection (*Shields et al., 1999*); therefore, the decrease in the area lacking MBP expression in the spinal cord 2 weeks after LPC injection indicates the success of remyelination (*Kuroda et al., 2017*). Immunohistochemical analysis revealed that mice that had received anti-CD41 mAb had a larger MBP-negative area than control mice (*Figure 3B,C*), indicating that platelet depletion inhibited remyelination. Myelin formation in the intact spinal cord was not changed after anti-CD41 mAb administration (*Figure 3—figure supplement 1D,E*). Electron microscopic analysis revealed that the thickness of myelin sheaths was significantly lower in mice treated with anti-CD41 mAb than in controls (*Figure 3D,E*), indicating that our immunohistological observations were consistent with the promotion of structural remyelination. We then asked whether TGF-β1 makes a major contribution to circulating factor-mediated remyelination. We used neutralizing TGF-β antibodies (*Cohn et al., 2007*; *Bohlen et al., 2017*); *Wang et al., 2018*) and intraperitoneally administrated the antibodies

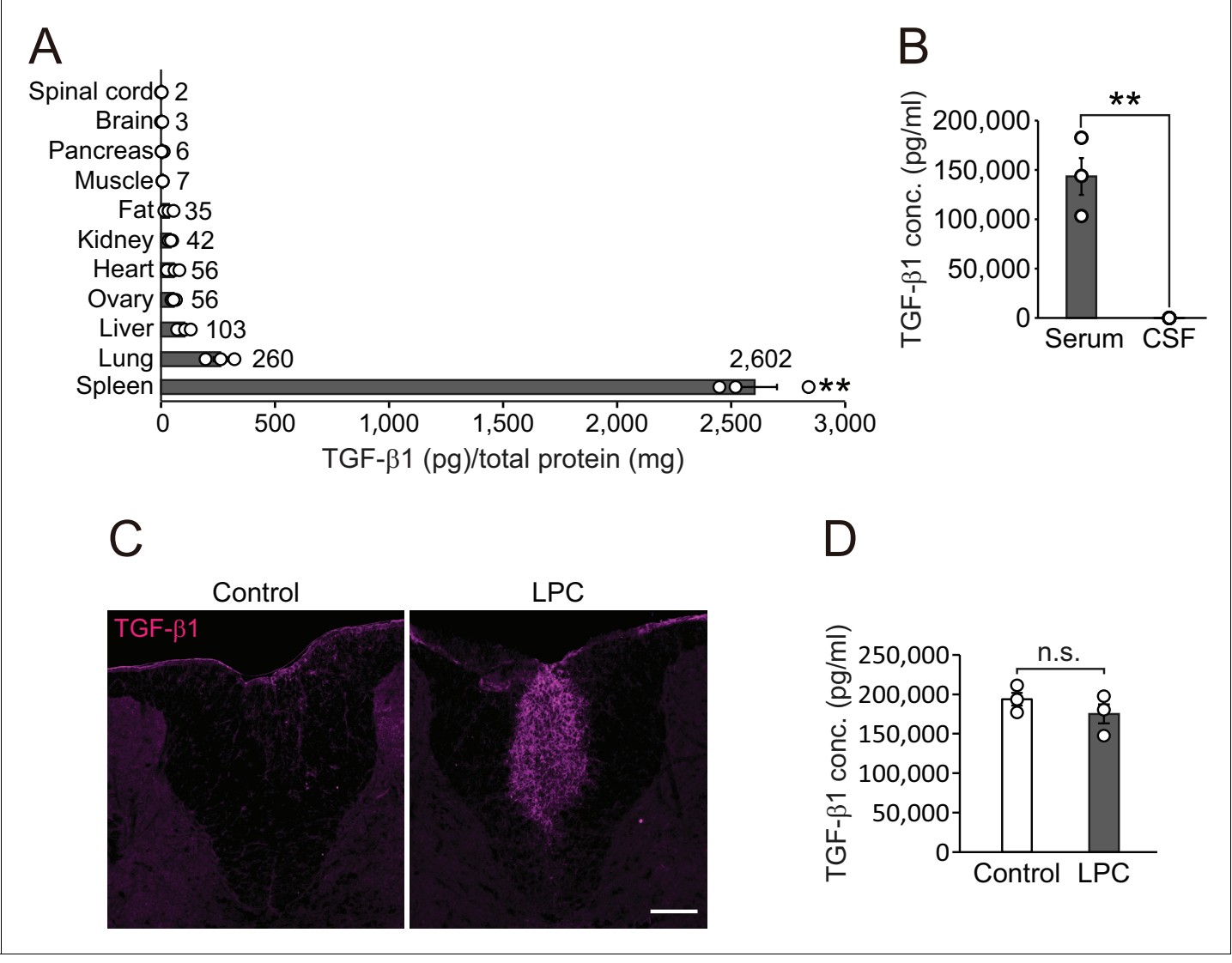

**Figure 2.** TGF-β1 level in circulation is higher than that in the CNS. (A) Quantification of TGF-β1 protein level in individual organs in intact mice (n = 3 for each), p<0.0001. (B) Quantification of TGF-β1 protein concentration in the serum and CSF (n = 3 for serum, n = 4 for CSF), p=0.0007. (C) Representative image of spinal cord section which is labeled with TGF-β1. The spinal cord sections were obtained 7 days after LPC injection. (D) Quantification of TGF-β1 protein concentration in the serum 7 days after LPC injection (n = 3 for each). NS indicates not significant difference. **p<0.01, Student's *t*-test or ANOVA with Tukey's post-hoc test. Error bars represent SEM. Scale bars represent 100 μm.

DOI: https://doi.org/10.7554/eLife.41869.005

into the mice every 2 days from 7 days after LPC injection. Mice injected with neutralizing TGF-β antibodies had a larger MBP-negative area around the LPC lesion than controls (*Figure 3F,G*), and the inhibition of remyelination following TGF-β neutralization was comparable to the level observed in platelet-depletion experiments ($P_{int}$ = 0.6126, single regression analysis). These data indicate that spontaneous remyelination depends on circulating TGF-β1.

## TGF-βRI in oligodendrocytes is involved in spontaneous remyelination

Because we detected high level of TGF-βRI expression in APC-labeled oligodendrocytes compared with that in platelet-derived growth factor alpha (PDGFRα)-labeled oligodendrocyte precursor cells (*Figure 4A*, *Figure 4—figure supplement 1A*), we hypothesized that circulating TGF-β1 acts directly on oligodendrocytes resulting in the promotion of remyelination. To investigate this, we generated inducible, oligodendrocyte-specific TGF-βRI knockout mice by using Plp-creER$^T$: TGF-βRI floxed

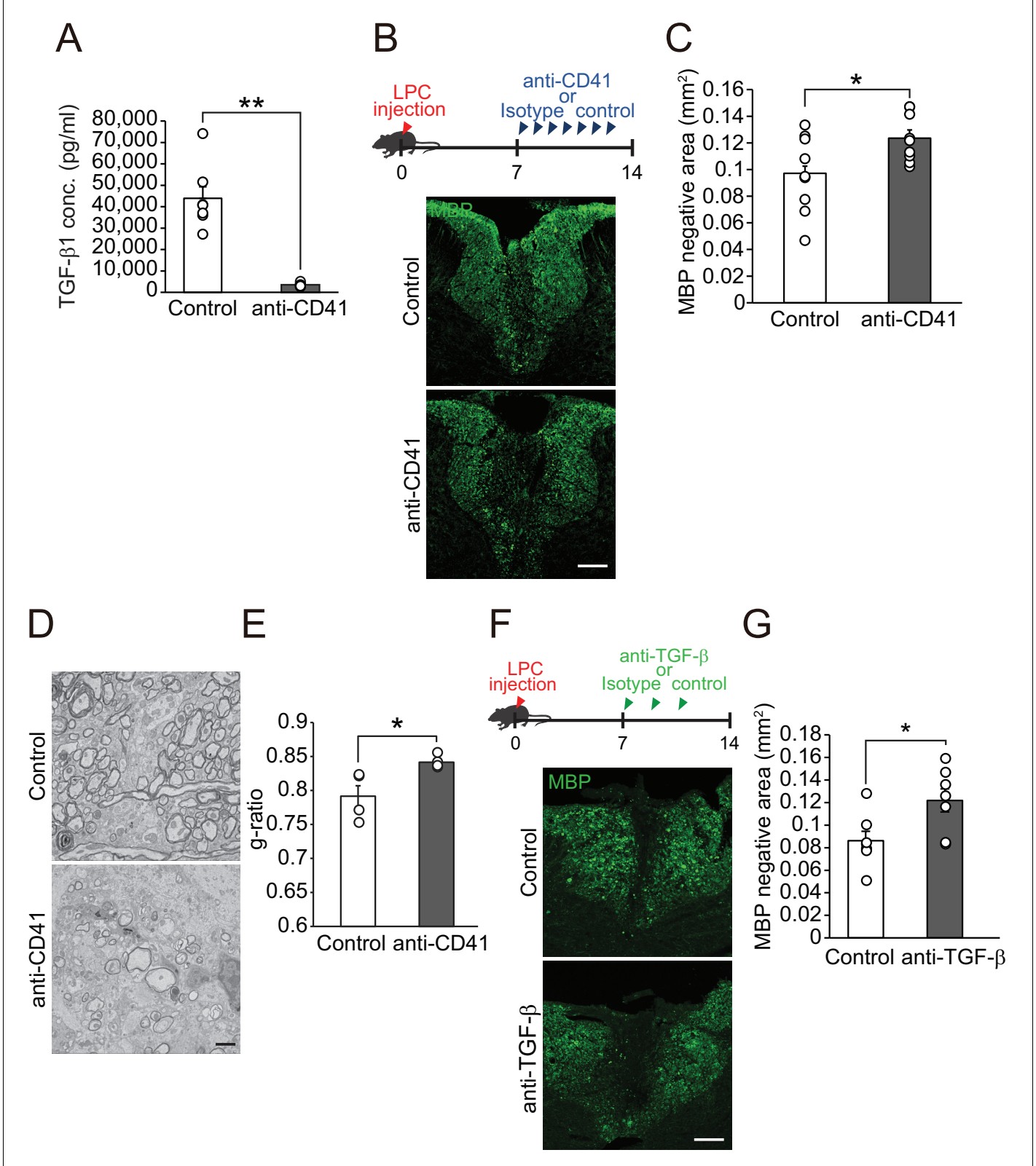

**Figure 3.** Circulating TGF-β1 supports spontaneous remyelination. (**A**) Quantification of circulating TGF-β1 level in the mice with platelet depletion. Plasma from mice were obtained from 7 days after the start of anti-CD41 mAbs injection (n = 7 for control IgG, n = 8 for anti-CD41 mAbs), p=0.0001. (**B**) Representative images of spinal cord sections labeled for MBP. Spinal cords were dissected from mice 14 days after LPC injection. (**C**) Quantification of an MBP-negative area in the dorsal column of spinal cord as indicated in B (n = 9 for each), p=0.0347. (**D**) Representative electron micrographs of
*Figure 3 continued on next page*

*Figure 3 continued*

myelin in the spinal cord. Sections were obtained from mice treated with anti-CD41 mAb 14 days after LPC injection. (E) Graphs show quantitation of the g-ratio indicated in D (n = 4 for each), p=0.0347. (F) Representative images of spinal cord sections labeled for MBP. Spinal cords were obtained 14 days after LPC injection. Injection of TGF-β neutralizing antibodies was started 7 days after LPC and continued every other day (n = 7), p=0.0278. (G) Quantification of an MBP-negative area in the dorsal column of spinal cord, as indicated in F. NS indicates not significant difference. **p<0.01, *p<0.05, Student's *t*-test. Error bars represent SEM. Scale bars represent 100 μm for B, F, 2 μm for D.

DOI: https://doi.org/10.7554/eLife.41869.006

The following figure supplement is available for figure 3:

**Figure supplement 1.** Circulating TGF-β leaks into the spinal cord after LPC injection.

DOI: https://doi.org/10.7554/eLife.41869.007

mice, which allowed tamoxifen-inducible deletion of TGF-βRI expression in oligodendrocytes (*Figure 4—figure supplement 1B,C*). Histological analysis revealed that the conditional compared with the control littermates, knockout mice had a larger MBP-negative area in the dorsal spinal cord (*Figure 4B,C*). LPC injection into the spinal cord caused motor deficits, which, however, improved over time due to spontaneous remyelination (*Hamaguchi et al., 2017*). We then asked whether TGF-βRI signaling is required for motor recovery after LPC injection. By behavioral analysis, significant inhibition of motor recovery was detected in the conditional knockout mice than in the control mice (*Figure 4D*). TGF-β1 treatment did not change the MBP-negative area in the oligodendrocyte-specific TGF-βRI knockout mice 2 weeks after LPC injection (*Figure 4E,F*). There was no significant difference in MBP-positive area in oligodendrocyte-specific TGF-βRI knockout mice that did not receive LPC injection (*Figure 4—figure supplement 1D,E*). These data indicate that direct interaction of circulating factors with CNS oligodendrocytes mediates spontaneous remyelination.

## TGF-β administration promotes CNS remyelination

To address whether TGF-β1 is sufficient to cause circulating-factor-mediated remyelination, we examined increase of remyelination efficiency by exogenous TGF-β1 administration in mice which did not show severe vascular barrier disruption. In the cuprizone feeding demyelination model, spontaneous remyelination occurs after terminating the administration of cuprizone; however, the remyelination efficiency is known to be lower when compared with the vascular barrier disruption demyelination model (*Kuroda et al., 2017*). Therefore, we asked whether intracerebroventricular infusion of TGF-β1 could promote remyelination after cuprizone diet. Histological analysis revealed that the MBP-negative area was smaller in the corpus callosum of mice administered with TGF-β1 than of control animals (*Figure 5A,B*). These data support the fact that exogenous TGF-β1 administration mimics remyelination with the same efficiency as that driven by circulating factors.

We then sought to determine the therapeutic potential of our results described above. Myelin oligodendrocyte glycoprotein (MOG) $_{35-55}$-induced EAE model is relatively controlled and shows demyelination pathology of MS (*Najm et al., 2015*). We started TGF-β1 administration at the peak of the clinical score (day 15) in EAE and continued administration during the observation period. TGF-β1-treated mice showed restoration of MBP expression in the spinal cord, whereas vehicle-treated mice showed sustained areas of white matter disruption (*Figure 5C,D*). By contrast, we observed no change in the accumulation of inflammatory cells (*Figure 5—figure supplement 1A,B*), cytokine release (*Figure 5—figure supplement 1C*), or cell proliferation (*Figure 5—figure supplement 1D*), indicating that TGF-β1-mediated remyelination does not depend on regulation of immunological responses. Consistent with these histological changes, behavioral evaluation showed that TGF-β1 administration starting 15 days after immunization improved neurological function significantly (*Figure 5E*). We also detected an intravenously injected fluorescently-labeled TGF-β1 in spinal cord tissues indicating that circulating TGF-β1 leaks into the CNS in EAE (*Figure 5F,G*). These data suggest that enhancement of circulating-factor activity is useful in treating demyelination.

## TGF-β enhanced human oligodendrocyte maturation

We then determined if our findings in mouse studies were translatable to humans. Immunohistochemical analysis of autopsy samples from MS patients revealed TGF-βRI expression in APC-positive oligodendrocytes (*Figure 6A*). We also investigated whether TGF-β1 would promote oligodendrocyte maturation in human samples. For this purpose, we used commercially available OPC cultures

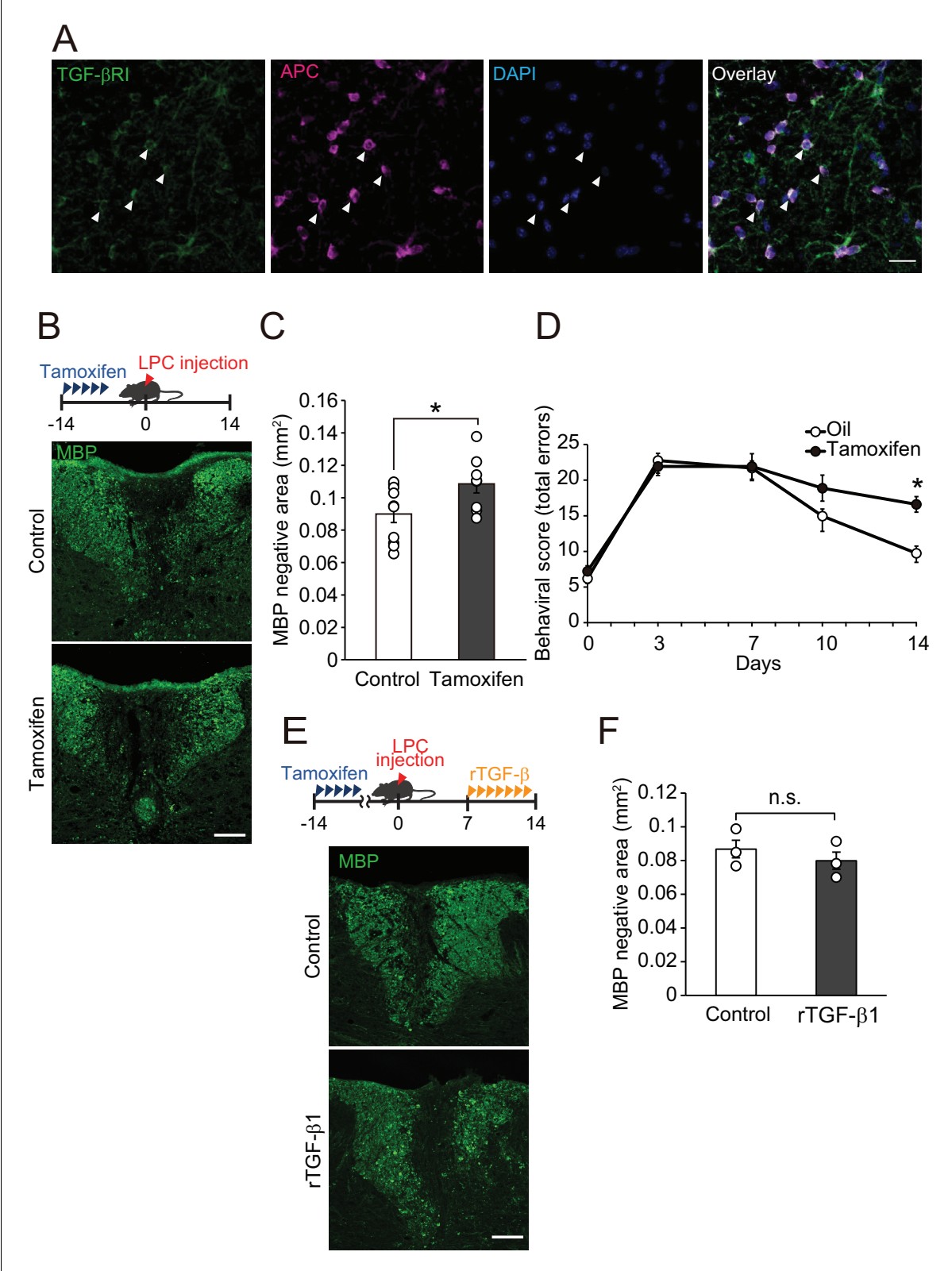

**Figure 4.** TGF-βRI in oligodendrocytes is required for remyelination. (A) Representative images of spinal cord sections double-labeled for APC and TGF-βRI. Spinal cords were obtained from control mice. The white arrowheads show APC⁺ TGF-βRI⁺ cells. (B) Representative images of spinal cord sections labeled for MBP. Spinal cords were obtained from the tamoxifen-treated conditional knockout mice 14 days after LPC injection. Control data was obtained from the conditional knockout mice without tamoxifen treatment. (C) Quantification of MBP-negative area in the dorsal column of spinal

*Figure 4 continued on next page*

*Figure 4 continued*

cord as indicated in B (n = 10 for control, n = 8 for conditional knockout mice), p=0.0364. (D) Motor function was assessed by beam walk test (n = 10 for each), p=0.0156. (E) Representative images of spinal cord sections labeled for MBP. Spinal cords were obtained from tamoxifen-treated conditional knockout mice 14 days after LPC injection. Data were obtained from conditional knockout mice with or without recombinant TGF-β1 treatment. (F) Quantification of MBP-negative area in the dorsal column of spinal cord, as indicated in D (n = 3 for each; p=0.4816). (F) *p<0.05, Student's *t*-test or Two-way ANOVA with Bonferroni's post-hoc test. Error bars represent SEM. Scale bars represent 20 μm for A, and 100 μm for B and D.
DOI: https://doi.org/10.7554/eLife.41869.008

The following figure supplement is available for figure 4:

**Figure supplement 1.** TGF-βRI is expressed in oligodendrocytes.
DOI: https://doi.org/10.7554/eLife.41869.009

established from human embryonic stem cells and treated the cells with recombinant human TGF-β 1. Real-time PCR analysis revealed that treatment with recombinant human TGF-β1 increased the levels of mRNAs encoding myelin-associated proteins (MBP, MAG, and PLP) in human oligodendrocyte cultures in vitro (*Figure 6B*), implying that TGF-β1–mediated oligodendrocyte maturation could also occur in humans.

## Discussion

The CNS is isolated from the peripheral environment under normal physiological conditions by the presence of vascular barrier, but this barrier is impaired in several types of diseases including AD, dementia, traumatic brain injury, and MS. Because disruption of the vascular barrier leads to hypoperfusion and causes excessive inflammatory response in acute CNS injury (*Zlokovic, 2008*), vascular impairment is increasingly thought to be associated with disturbances in CNS homeostasis. In contrast, chronic demyelinating lesions are characterized by less damage of blood brain barrier (BBB) when compared with the damage caused by an active lesion (*Kirk et al., 2003*; *Lassmann et al., 1994*). Consistent with this, an area of chronic demyelination that fails to undergo remyelination is considered to arise due to the decline of maturation activity in oligodendrocytes (*Chang et al., 2002*; *Wolswijk, 1998*). Although it is not clear why remyelination fails to occur in chronic demyelination (*Franklin, 2002*), the relative kinetics of vascular barrier disruption and remyelination suggest that it is possible that peripheral tissue–derived factors play an important supportive role during remyelination when the integrity of the BBB is impaired, such as during the remission phase of MS.

In this context, we identified circulating TGF-β1 as a candidate factor that promotes oligodendrocyte maturation. TGF-β1 does not cross the intact BBB (*Kasten et al., 2003*) and TGF-β1 concentration in the CSF is maintained at a lower level than that which is in the plasma of a healthy subject (*Tarkowski et al., 2002*). However, it is important to note that TGF-β1 is also expressed in CNS cells, including astrocytes. In MS, TGF-β1 expression is detected in the lesions (*De Groot et al., 1999*), and TGF-β1 mRNA expression in the spinal cord increased in advance of remission of EAE (*Issazadeh et al., 1995*). Since astrocyte-derived TGF-β1 regulates the local inflammatory response which contributes to the initiation of EAE (*Luo et al., 2007*), CNS-cell-mediated TGF-β1 signaling may be involved in remyelination. However, TGF-β1 promotes Jagged1 expression in demyelinated lesions, and this upregulation inhibits oligodendrocyte maturation (*John et al., 2002*), opposing the pro-remyelination effect of circulating TGF-β1. In addition, we found that oligodendrocyte maturation was detected in cells that were cultured in the presence of TGF-β1 at a concentration 29-fold higher than that of the CSF (*Figure 1H,I*). These data support our hypothesis that circulating TGF-β1 enters the CNS, and potentially contributes to remyelination.

This study resolves a long-standing debate about the therapeutic mechanism by which systemic TGF-β1 acts in the treatment of EAE and reveals the capacity of circulating TGF-β1 to contribute to the regeneration of neuronal networks after injury. Our study is currently limited to remyelination in MS. However, CNS regeneration is essential for efficient functional recovery after inflammation and injury in general. Therefore, our study predicts a new direction in CNS regeneration research with respect to the contribution of circulating factor in regeneration, which is supported by the previous findings that plastic changes of neuronal networks depend upon the systemic milieu (*Ruckh et al., 2012*; *Villeda et al., 2011*; *Villeda et al., 2014*). In other contextual studies, it has been suggested that levels of circulating Growth differentiation factor 11, which belongs to the TGF-β family, decline

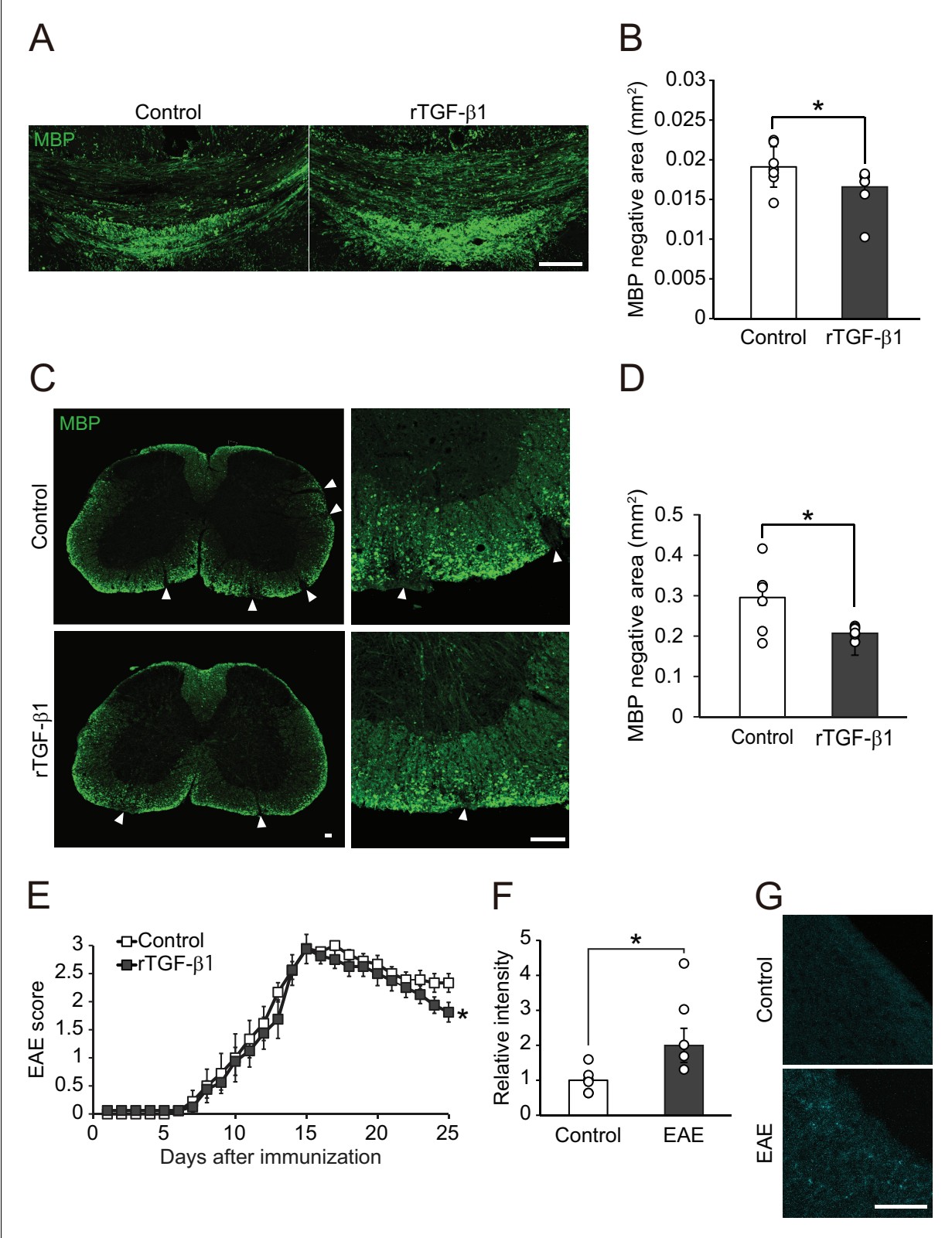

**Figure 5.** TGF-β1 treatment promotes remyelination. (**A**) Representative image of brain section labeled for MBP. Brains were obtained from the cuprizone fed mice, along with TGF-β1 administration for 14 days after removal of cuprizone diet. (**B**) Quantification of MBP-negative area in the corpus callosum as indicated in A (n = 7 for control, n = 8 for TGF-β1), p=0.0444. (**C**) Representative image of spinal cord section labeled for MBP. Spinal cords were dissected from the mice 25 days after EAE induction. Right panels show high magnification of left panels. (**D**) Quantification of MBP-negative area

*Figure 5 continued on next page*

*Figure 5 continued*

in the spinal cord as indicated in C (n = 7 for each), p=0.0125. (**E**) EAE score of the mice with TGF-β1 administration. TGF-β1 administration was started 15 days after EAE induction and continued daily injection at the end of period (n = 13 for control, n = 12 for TGF-β1), p=0.049. (**F**) Relative intensity of fluorescent dye-labeled recombinant mouse TGF-β1 in the spinal cord obtained from EAE mice (n = 6 for control, n = 5 for EAE), p=0.0193. (**G**) Representative image of spinal cord section from EAE mice. EAE mice received fluorescent-dye-labeled recombinant mouse TGF-β1. *p<0.05, Student's *t*-test or two-way ANOVA with Bonferroni's post-hoc test. Error bars represent SEM. Scale bars represent 100 μm.

DOI: https://doi.org/10.7554/eLife.41869.010

The following figure supplement is available for figure 5:

**Figure supplement 1.** TGF-β treatment does not regulate accumulation of inflammatory cells.

DOI: https://doi.org/10.7554/eLife.41869.011

with age and is linked to age-related systemic impairments such as cardiac hypertrophy (*Loffredo et al., 2013*), skeletal muscle dysfunction (*Sinha et al., 2014*), and decline of neurogenesis in the subventricular zone (*Katsimpardi et al., 2014*). These studies emphasize that circulating factors should be regarded as important molecular regulators of mammalian aging and have potentially broad-reaching implications.

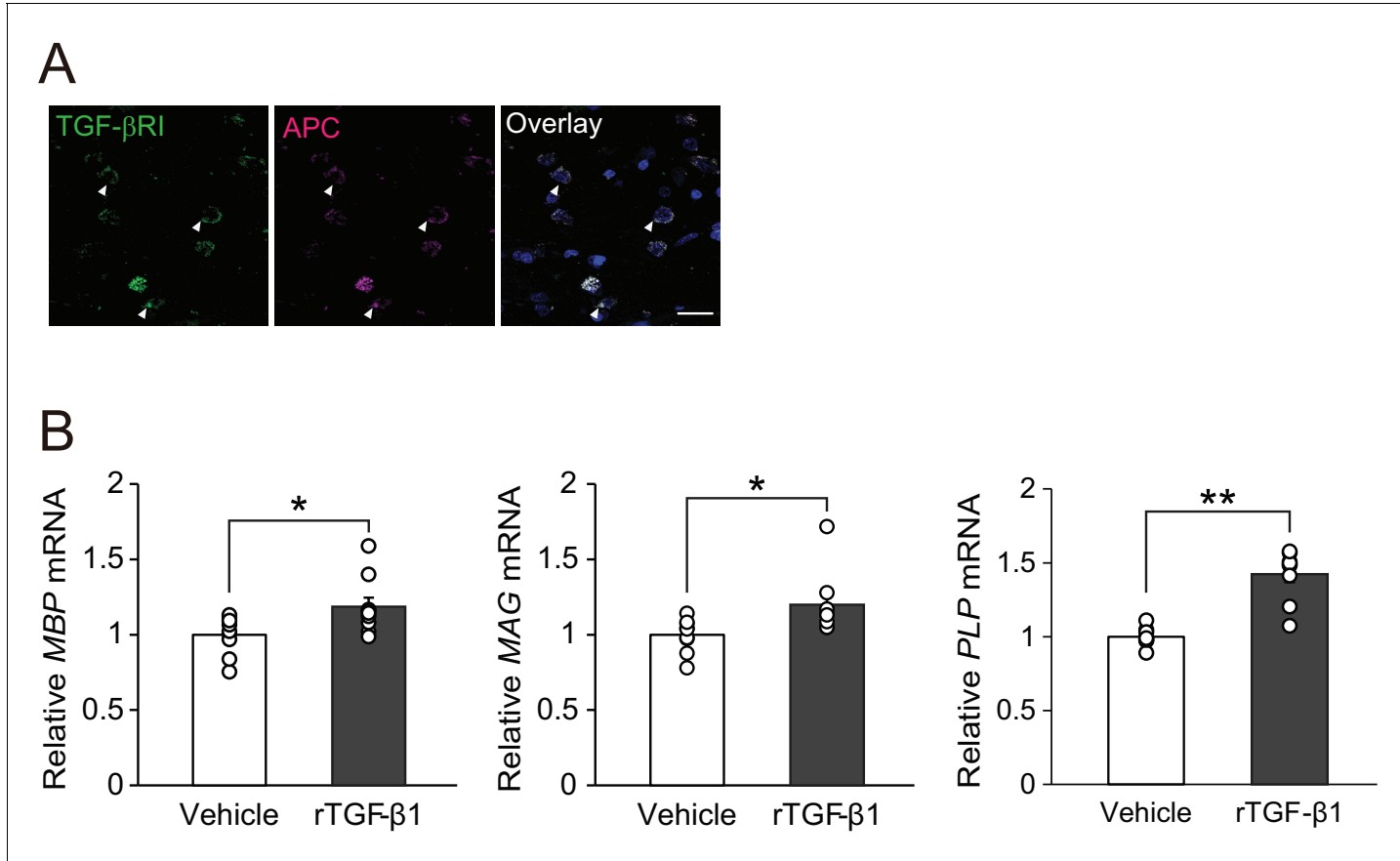

**Figure 6.** TGF-β1 stimulates human oligodendrocyte maturation. (**A**) Representative images of spinal cord sections double-labeled for APC with TGF-β RI. Brain samples were obtained post-mortem from MS patients. (**B**) Relative expression of indicated mRNA in human oligodendrocyte. The cells were treated with recombinant human TGF-β1 (10 nM) for 3 days (n = 9 for each), p=0.0278 (*MBP*), 0.0222 (*MAG*), 0.0001 (*PLP*). **p<0.01, *p<0.05, Student's *t*-test. Error bars represent SEM. Scale bar represents 50 μm.

DOI: https://doi.org/10.7554/eLife.41869.012

# Materials and methods

## Key resources table

| Designation | Source or reference | Identifiers | Additional information |
|---|---|---|---|
| Tg(Plp1-cre/ERT)3Pop | PMID:12481300 | RRID:MGI:3696409 | |
| ALK5 flox | PMID:12842983 | | Dr. Stefan Karlsson (Lund University) |
| Myelin Basic Protein antibody | PMID:20209960 | RRID:AB_305869 | IHC (1:500) |
| Goat Anti-Human Olig2 Polyclonal antibody | PMID:18615534 | RRID:AB_2157554 | IHC (1:300) |
| Anti-APC (Ab-7) Mouse mAb (CC-1) antibody | PMID:16705673 | RRID:AB_2057371 | IHC (1:500) |
| Donkey Anti-Mouse IgG (H + L) Antibody, Alexa Fluor 594 Conjugated | PMID:23970788 | RRID:AB_141633 | IHC (1:500) |
| Donkey Anti-Goat IgG (H + L) Antibody, Alexa Fluor 647 Conjugated | PMID:25505326 | RRID:AB_141844 | IHC (1:500) |
| Phospho-Smad2 (Ser465/467)/Smad3 (Ser423/425) (D27F4) Rabbit mAb antibody | PMID:28324027 | RRID:AB_2631089 | WB (1:1000) |
| Smad2/3 (D7G7) XP Rabbit mAb antibody | PMID:29056512 | RRID:AB_10889933 | WB (1:1000) |
| Anti-Rabbit IgG Phototope-HRP Western Blot Detection System Antibody, Horseradish Peroxidase Conjugated | | RRID:AB_2099234 | WB (1:2000) |
| Rat Anti-CD41 Monoclonal Antibody, Unconjugated, Clone MWReg30 | PMID:30590031 | RRID:AB_395084 | |
| Goat Anti-Mouse Pdgf r alpha Polyclonal antibody, Unconjugated | PMID:28111199 | RRID:AB_2236897 | IHC (1:500) |
| TGFbeta RI (V-22) antibody | PMID:29161592 | RRID:AB_632493 | IHC (1:50) |
| Donkey Anti-Rat IgG (H + L) Antibody, Alexa Fluor 488 Conjugated | PMID:28280459 | RRID:AB_141709 | IHC (1:500) |
| TGF-beta 1,–2,−3 MAb (Clone 1D11) antibody | PMID: 28521131 | RRID:AB_357931 | |
| Q-Plex Mouse Cytokine - Screen (16-Plex) | Quansys Biosciences | Cat. #110949 MS | |
| Human Oligodendrocyte differentiation Kit | Millipore | Cat. #CS204496 | |
| Human recombinant TGF-β1 | R and D Systems | Cat. #240-B | |
| Recombinant Murine basic FGF | Peprotech | Cat. #450–33 | |
| Recombinant Murine PDGF-AA | Peprotech | Cat. #315–17 | |

*Continued on next page*

*Continued*

| Designation | Source or reference | Identifiers | Additional information |
|---|---|---|---|
| LY364947 | Calbiochem | Cat. #616451 | |
| Recombinant mouse TGF-b1 | R and D Systems | Cat. #7666 MB-005 | |
| Mouse/Rat/Porcine/ Canine TGFb1 Quantikine ELISA | R and D Systems | Cat. #MB100B | |
| Cell Proliferation ELISA, BrdU (colorimetric) | Sigma Aldrich | Cat. #11647229001 | |
| L-α-Lysophosphatidylcholine (LPC) from bovine brain | Sigma Aldrich | Cat. #L1381 | |
| InhibitorSelect 384-Well Protein Kinase Inhibitor Library I | Calbiochem | Cat. #539743 | |

## Mice

All experimental procedures were approved by the Institutional Animal Care Committee of Osaka University (no. 24-067-055) and the Committee on the Ethics of Animal Experiments of the National Institutes of Neuroscience, National Center of Neurology and Psychiatry (no. 2018042R5). C57BL/6J mice were obtained from Charles River Japan or Japan SLC. Plp-CreERT mice (C57BL/6 background) were purchased from the Jackson Laboratory. TGF-βRI (ALK5) floxed mice (C57BL/6 background) (*Larsson et al., 2003*) were kindly provided by Prof. Stefan Karlsson (Lund University). Mice were born and held in the specific pathogen free (SPF) conditions. Mice were housed in an air-conditioned room at $23 \pm 1°C$ with a 12 hr light–dark cycle and had free access to water and food.

Oligodendrocyte-specific TGFβRI deletion mice were obtained by crossing the TGFβRI floxed mice with the Plp-CreERT mice. Cre recombination in the mice thus generated was induced by administering 4-hydroxytamoxifen (1 mg/kg/day, i.p.; Sigma-Aldrich, dissolved in dimethylsulphoxide (DMSO): ethanol: corn oil [4:6:90] mixture) daily over five consecutive days. Tamoxifen administration was started 7 days before LPC injection. TGF-βRI deletion in APC-labeled oligodendrocytes was immunohistochemically confirmed on the spinal cord tissues 14 days after LPC injection. In all cases, the experimenter was blinded to the origin of the samples. Mice were randomly allocated into groups.

## Serum and plasma collection

Cardiac blood was collected from 8 week old female C57BL/6J mice. For serum preparation, blood was collected and incubated for 30 min at room temperature, and then samples were centrifuged at $2000 \times g$ for 15 min. The supernatant (serum) was collected and stored at −80°C. For plasma preparation, blood was collected using a heparin coated capillary (TERUMO) or an EDTA coated capillary (Vitrex Medical A/S). Samples were centrifuged at $2000 \times g$ for 15 min. The supernatant (plasma) was collected and stored at −80°C.

For digestion experiments, serum was incubated at 37°C for 2 hr with 50 μg/ml DNase (Sigma, DN25) or 1 μg/ml RNase (Roche) at 37°C for 1 hr. For heat treatment, the serum was heated at 95°C for 5 min.

## Primary culture of oligodendrocytes

Oligodendrocytes were obtained from postnatal day 1 mice. The cerebral cortices were dissected in phosphate buffer saline (PBS) and dissociated into single-cell suspensions using the 0.25% Trypsin-PBS by incubation at 37°C for 15 min. After neutralization by Dulbecco's modified Eagles medium (DMEM) containing 10% fetal bovine serum (FBS), cells were centrifuged at $300 \times g$ for 5 min, suspended in 10% FBS-DMEM, and filtered through a 70-μm nylon cell strainer. Single cells were plated at a density of $3–6 \times 10^5$ cells/ml on poly-L-lysine (PLL)–coated dishes (Greiner Bio-One) and maintained at 37°C with 7% $CO_2$ in 10% FBS-DMEM. Ten days after culturing, cells were washed in PBS. The remaining cells were treated with 0.05% Trypsin-PBS at 35°C for 4 min, and then tapped gently. The detached cells were filtered through a 40 μm nylon cell strainer and plated into non-coated dishes. After a 30-min incubation at 37°C, non-adherent cells were collected and plated at a density

of $3 \times 10^4$ cells/well into PLL-coated 96-well plates in OPC medium. OPC medium was constituted as follows: DMEM contained 4 mM L-glutamine (Sigma), 1 mM sodium pyruvate (Sigma), 0.1% bovine serum albumin (BSA; minimum 98% electrophoresis grade, Sigma), 50 µg/ml apo-transferrin (Sigma), 5 µg/ml insulin (Sigma), 30 nM sodium selenite (Sigma), 10 nM biotin (Sigma), 10 nM hydrocortisone (Sigma), 10 ng/ml platelet-derived growth factor-AA (PDGF-AA; Pepro Tech), and 10 ng/ml basic fibroblast growth factor (basic-FGF, Pepro Tech). Immunocytochemistry revealed that 58.1 ± 0.9% of the cells in the culture were co-labeled with Olig2, an oligodendrocyte marker (data not shown).

After 3 days of culturing, we performed pharmacological screening. The following drugs were used: Inhibitor Select 384-well Protein Kinase Inhibitory Library I (1:1000, Calbiochem), LY364947 (a transforming growth factor [TGF]-β receptor I [TGF-βRI] kinase inhibitor) (1 µM, Calbiochem), and recombinant mouse TGF-β1 (0.1–10 ng/ml, R and D Systems). Cells were cultured for an additional 5 days and used for evaluation in a differentiation assay.

## siRNA transfections

Mouse TGF-βRI siRNA (ID: s75059) were purchased from Ambion. Transfection of cultured oligodendrocytes with TGF-βRI siRNA was performed using Lipofectamine RNAiMAX (Invitrogen). Cells were lysed 3 days after transfection and evaluated the TGF-βRI mRNA level by real-time PCR.

## Immunocytochemistry

Cells were fixed with 4% paraformaldehyde (PFA) in PBS for 30 min at room temperature, followed by blocking with PBS containing 5% bovine serum albumin (BSA; minimum 98% electrophoresis grade, Sigma-Aldrich) and 0.1% Triton X-100 for 1 hr at room temperature. The cells were incubated with primary antibodies diluted in the blocking solution (PBS containing 5% BSA and 0.1% Triton X-100) overnight at 4°C. The following antibodies were used for primary antibodies: rat anti-myelin basic protein (MBP; 1:500, Abcam, AB7349), goat anti-Olig2 antibody (1:300, R and D Systems, AF2418), and mouse anti-mouse APC (ab-7) (CC1; 1:500, Calbiochem, OP80). As secondary antibodies, the cells were incubated for 1 hr at room temperature with Alexa Fluor 488–conjugated donkey antibody against rat IgG, Alexa Fluor 594–conjugated donkey antibody against mouse IgG, or Alexa Fluor 647–conjugated donkey antibody against goat IgG (1:500, Invitrogen). The nuclei were stained with 4',6-Diamidino-2-Phenylindole (DAPI, 1 µg/ml, Dojindo Laboratories) for 10 min. Images were acquired by fluorescence (Olympus BX53, 44FL).

To evaluate oligodendrocyte maturation, images were acquired with an IN Cell Analyzer 6000 (GE Healthcare) and quantified MBP$^+$ area/Olig2$^+$ cells to estimate oligodendrocyte differentiation using IN Cell developer (GE Healthcare).

## Western blot analysis

Cells were homogenized in 10 mM Tris-HCl (pH 7.4), 150 mM NaCl, 1% Triton-X 100, and 1 mM ethylenediaminetetraacetic acid (EDTA) containing protease inhibitor (Roche). The lysates were clarified by centrifugation at 8000 $g$ at 4°C for 20 min, and the supernatants were collected and normalized for protein concentration. Proteins were separated by 10% sodium dodecyl sulfate–polyacrylamide gel electrophoresis (SDS-PAGE) and transferred onto polyvinylidene difluoride membranes (Immobilon-P, Millipore). After blocking with PBS containing 5% skim milk and 0.05% Tween 20, the membranes were incubated with primary antibodies overnight at 4°C, followed by incubation with a fluorescently-labeled secondary antibody for 1 hr at room temperature. The following antibodies were used: rabbit anti-phospho-Smad2 (Ser465/467)/Smad3 (Ser423/425) (8828, Cell Signaling Technology) and rabbit anti-Smad2/3 (D7G7) (8685, Cell Signaling Technology). Horseradish peroxidase–conjugated anti-rabbit IgG antibody was used as secondary antibody (Cell Signaling Technology).

Immunoreactive bands were detected using a fluorescence-conjugated secondary antibody and an enhanced chemiluminescence (ECL) system (WBKLS0100, Millipore), and visualized on a LAS-4000 imaging system (Fujifilm). The protein bands were quantified using the ImageJ software.

## Enzyme-linked immunosorbent assay (ELISA)

TGFβ levels in mouse serum, cerebrospinal fluid (CSF), or tissue lysates were examined using a Mouse/Rat/Porcine/Canine TGFβ1 Quantikine ELISA (R and D Systems). CSF was collected by

cisterna magna puncture using 29-gauge needle. Mouse tissues were lysed in 5 mM Tris-HCl (pH 8.0), 150 mM NaCl, 0.02% sodium aside, 0.1% Triton-X, and protease inhibitor (Complete; Roche).

## Surgical procedure

Female mice (8–10 weeks old) were anesthetized with a mixture of Dormicam, 4 mg/kg; Vetorphale, 5 mg/kg; Domitor 4 mg/kg. The mice underwent laminectomy at Th12 and received injections of 2 µl of 1% (w/v) LPC dissolved in PBS into the dorsal column midline at a depth of 0.5 mm.

For administration of pharmacological reagent, an Alzet osmotic pump (model no. 1002; Alzet Corp) was filled with LY364847 (7.25 µg/kg of body weight per day). The pump was connected to a delivery tube, which was placed close to the site of the lesion 3 days after LPC injection. The pump was implanted subcutaneously on the back of the animal.

For administration of TGF-β neutralizing antibodies, TGF-beta 1, 2, 3 Antibody (R and D system) was intraperitoneally injected every 2 days (10 mg/kg of body weight per one injection). Administration of antibodies was stared 7 days after LPC injection. Control IgG was used for control experiments.

## Platelet depletion

Mice were intraperitoneally administered with rat anti-CD41 mAbs (BD Biosciences, 553847) or rat IgG1 control (BD Biosciences, 553922) for seven consecutive days from 7 days after LPC injection (10 µg/mouse on the first day, followed by 5 µg/mouse on following days). Platelet depletion was confirmed by flow cytometry with an FITC rat anti-mouse CD41 antibody (1:200, BD Pharmingen, 553848) on BD FACS Verse instrument.

## Demyelination by cuprizone diet

Mice were fed with 0.2% (w/w) cuprizone (Sigma, C9012) for 12 weeks and then returned to a normal diet. Mice were maintained in sterile, pathogen-free conditions. For intraventricular administration of recombinant TGF-β1, an Alzet osmotic pump (model no. 1002; Alzet Corp) was filled with recombinant mouse TGF-β1 (266.5 µg/kg of body weight per day) and stereotactically equipped with a cannula (Brain Infusion Kit 3, ALZET Cupertino, 0008851) targeting the lateral ventricle of the brain (coordinate to bregma: anterior, 0.5 mm, lateral, −1.1 mm; ventral, 2.5 mm).

## Histological analysis

Mice were transcardially perfused with PBS followed by 4% PFA in PBS. Brain and spinal cord were post-fixed with 4% PFA in PBS overnight at 4°C following immersion in 30% sucrose in PBS. Tissues were embedded in optimal cutting temperature compound (Tissue-Tek, Sakura Finetek USA Inc), and then 30 µm sections were cut and mounted on Matsunami adhesive silane-coated slides (Matsunami Glass). The sections were permeabilized with PBS containing 0.1% Triton X-100% and 3% normal donkey serum for 1 hr at room temperature. The sections were then incubated with primary antibodies overnight at 4°C, and then incubated with fluorescently labeled secondary antibodies for 1 hr at room temperature. The primary antibodies used were as follows: goat anti-mouse PDGFRα (1:500, R and D Systems, AF1062), rabbit anti-human TGF-βRI (V-22) (1:50, Santa Cruz Biotechnology, SC-398), mouse anti-mouse APC (ab-7) (CC1; 1:500, Calbiochem, OP80), rat anti-myelin basic protein (MBP; 1:500, Abcam, AB7349). Secondary antibodies were Alexa Fluor 488-, 568-, and 594-conjugated antibodies produced in donkey (1:500 in PBS with 0.05% Tween-20 [Sigma]; Invitrogen). Nuclear staining was performed with 1 µg/ml DAPI. Images were acquired by fluorescence (Olympus BX53, 44FL) or confocal laser-scanning microscopy (Olympus FluoView FV1200).

To evaluate remyelination, MBP negative area in the dorsal column of the spinal cord was measured by Image J software. The mean was calculated from at least 10 sections spaced 100 µm apart.

To evaluate inflammation index, sections were stained by hematoxylin and eosin. The number of inflammatory foci, which reflects an accumulation of immune cells, were counted in the sections and were normalized to the area per $mm^2$.

## Electron microscopy

Mice were transcardially perfused with ice-cold PBS followed by a fixative (2% paraformaldehyde in PBS). Spinal cord tissues were removed and postfixed in the same fixative at 4°C overnight. Spinal

cords were sliced into 50 µm pieces using a Vibrating Blade Microtome and washed in 0.1 M PB to prepare the sample for electron microscopy at Hanaichi UltraStructure Research Institute. Samples were observed by transmission electron microscopy (H-7650, Hitachi). The g-ratio was calculated by dividing the diameter of the inner axon by the diameter of the myelinated fiber.

### Beam walking test

Mice were pre-trained 2 days before test. A recording was made of mice walking on a wooden beam (1.05 m length, 10 mm width, Beam test experimental device [Brain Science Idea Co. Ltd., cat. no. BS-VAM]). Every footstep of each hindlimb was scored according to the following criteria: 0 = normal, no footslip; 1 = mild footslip, in which a part of foot is seen below the surface of the beam; 2 = severe footslip, in which whole foot is seen below the surface of the beam. Total scores were created by summing up the individual scores. The data was obtained by taking the average of three trials per beam per mouse.

### EAE induction

Female mice were immunized by subcutaneous injection of 200 µl emulsion (100 µg $MOG_{35-55}$ peptide [MEVGWYRSPFSRVVHLYRNGK, Sigma-Aldrich] in complete Freund's adjuvant emulsion [CFA, Difco] containing 500 µg *Mycobacterium tuberculosis H37Ra* [Difco]). Mice were injected 100 ng pertussis toxin (List Biological Laboratories) intravenously 0 hr and 48 hr after immunization. The mice were assessed for signs of EAE according to the following scale: 0, no clinical signs; 0.5, tail tip droop; 1, partially tail droop; 1.5, tail paralysis; 2.0, hindlimb weakness; 2.5, one hindlimb paralysis; 3.0, both hindlimb paralysis; 3.5, hindlimb paralysis and forelimb weakness; 4.0, hindlimb paralysis and one forelimb paralysis; 4.5, hindlimb paralysis and both forelimb paralysis; 5.0, moribund or death.

To investigate the therapeutic role of TGF-β1 on EAE, intravenous injection of recombinant TGF-β1 (5 µg/kg of body weight per day) was started at 15 days after immunization and continued for up to 10 additional days.

### Splenocyte culture

Splenocytes were collected from mouse spleen 21 days after EAE induction. Spleen were dissected and triturated by using 1-ml syringe and 100-µm nylon cell strainer in RPMI1640 (Sigma), and then cells were treated with hemolysis buffer (Immuno-Biological Laboratories). After centrifugation, cells were suspended in RPMI1640 containing 10 mM HEPES buffer (Sigma), 10% FBS, 50 µM 2-mercaptoethanol, and penicillin-streptomycin (Gibco). Cells were cells were plated at density of $1 \times 10^4$ cells/well into 96-well plates and were maintained at 37°C with 5% $CO_2$. To evaluate cell proliferation, cells were cultured with 20 µg/mL $MOG_{35-55}$ peptide for 72 hr. Cell proliferation was assessed using the Cell Proliferation ELISA and BrdU (colorimetric) kit (Sigma). BrdU solution was added into the culture 24 hr before the end of culturing. To evaluate the cytokine release, cells were cultured with 20 µg/mL $MOG_{35-55}$ peptide for 72 hr. The supernatants of the culture were collected and were used for the analysis of cytokine level by Q–Plex array (Quansys Biosciences).

### Human oligodendrocyte culture

Human oligodendrocytes (Human Oligodendrocyte differentiation Kit; Millipore, CS204496) were cultured in the Human OPC Expansion Complete Media (Millipore). After seven days of culture, the cells were re-plated at a density of $5 \times 10^4$ cells/well in 24-well plates in Human OPC Expansion Complete Media. Cells were treated with human recombinant TGF-β1 (0.1–10 ng/ml, R and D Systems) for 3 days. After culturing, the cells were used for real-time PCR analysis. Immunocytochemistry revealed that 97.68 ± 1.2% of DAPI$^+$ cells in the culture were co-labeled with Olig2, an oligodendrocyte marker (data not shown).

### Immunohistochemistry of human tissue

We obtained autopsied spinal cord tissues from four healthy individuals (three men; median age: 68 years; range: 63–70 years) and three individuals with multiple sclerosis (one man; median age: 66 years; range: 63–85 years). Formalin-fixed spinal cord samples were embedded in paraffin and cut into 10-µm-thick sections for immunohistochemistry. The sections were deparaffinized and treated

with 5% BSA and 0.3% Triton X-100 in PBS for 1 hr at room temperature. The sections were incubated with primary antibodies for 24 hr at 4°C and then incubated with secondary antibodies for 1 hr at room temperature. Primary antibodies were rabbit anti-human TGF-βRI (V-22) (1:50, Santa Cruz Biotechnology, SC-398) and mouse anti-APC (CC1; 1:20, Calbiochem). Secondary antibodies were Alexa Fluor 488- and 594- conjugated antibodies produced in donkey (1:500; Invitrogen). Nuclear staining was performed with 1 µg/ml DAPI. Images were acquired by confocal laser-scanning microscopy (Olympus FluoView FV1200). The research protocol was approved by the Human Use Review Committees of Toneyama National Hospital (TNH-2018031). Informed consent was obtained from all subjects.

## Quantitative RT-PCR

Total RNA was isolated using the RNeasy Mini kit (Qiagen), and cDNA was synthesized using the PrimeScript II High Fidelity RT-PCR Kit (Takara Bio Inc). cDNA fragments were amplified using the following primer pairs: mouse TGF-β1 forward, GGACTCTCCACCTGCAAGAC; mouse TGF-β1 reverse, GACTGGCGAGCCTTAGTTTG; mouse TGFβ receptor type I forward, TGCCATAACCGCACTGTCA; mouse TGFβ receptor type I reverse, AATGAAAGGGCGATCTAGTGATG; mouse GAPDH forward, TGTGTCCGTCGTGGATCTGA; mouse GAPDH reverse, TTGCTGTTGAAGTCGCAGGAG; human MBP forward, GGCCTTACCACTCGGTGATTAT; human MBP reverse, TTTCTGCAAAGGACTCTGTGAAGA; human PLP forward, CCATGCCTTCCAGTATGTCATC; human PLP reverse, GCCCTCAGCCAGCAGGA; human MOG forward, TTGTGTGAGTGCCTGGCAA; human MOG reverse, TGCCTCCACTCCGGTAATTG; human GAPDH forward, AGGGCTGCTTTTAACTCTGGT, human GAPDH reverse, CCCCACTTGATTTTGGAGGGA. Samples for SYBR Green assays consisted of a $1 \times$ final concentration of Power SYBR Green PCR Master Mix (Applied Biosystems, 4385612), 200 nM gene-specific primers, and 10 ng cDNA. PCR conditions included one cycle at 95°C for 10 min, followed by 40 cycles of 95°C for 15 s and 60°C for 60 s (ABI ViiA7 real-time PCR system; Applied Biosystems). A melting temperature analysis was carried out following PCR to monitor amplification specificity. Relative mRNA expression was normalized against GAPDH mRNA levels in the same samples and calculated by the Δ/Δ-Ct method.

## Statistical analysis

Data are presented as mean ± SEM. Statistical significance between groups was determined by unpaired Student's $t$-test, paired $t$-test, or repeated-measures ANOVA followed by post-hoc comparison with Dunnett's test or Tukey-Kramer test. $p < 0.05$ was considered to represent a significant difference.

## Acknowledgements

We are grateful to Prof. Kazuhiro Ikenaka, Dr. Takeshi Shimizu, and Dr. Fumiko Itoh for support conditional knockout mouse experiments. This work was supported by a Grant-in-Aid of Scientific Research (B) from the Japan Society for the Promotion of Science to RM (16H04672, 19H03554), Scientific Research (S) from the Japan Society for the Promotion of Science to TY (17H06178), and The Uehara Memorial Foundation to RM.

## Additional information

### Funding

| Funder | Grant reference number | Author |
|---|---|---|
| Japan Society for the Promotion of Science | 16H04672 | Rieko Muramatsu |
| The Uehara Memorial Foundation | 2017 | Rieko Muramatsu |
| Japan Society for the Promotion of Science | 17H06178 | Toshihide Yamashita |
| Japan Society for the Promotion of Science | 19H03554 | Rieko Muramatsu |

The funders had no role in study design, data collection and interpretation, or the decision to submit the work for publication.

## Author contributions

Machika Hamaguchi, Data curation, Formal analysis, Investigation, Writing—review and editing, Carried out almost of all experiments, Collected the data; Rieko Muramatsu, Conceptualization, Data curation, Formal analysis, Funding acquisition, Investigation, Methodology, Writing—original draft, Project administration, Writing—review and editing, Carried out western blot analysis and collected the data, Developed the concept, Designed experiments, Wrote the manuscript; Harutoshi Fujimura, Hideki Mochizuki, Resources, Writing—review and editing, Contributed human sample experiments; Hirotoshi Kataoka, Methodology, Writing—review and editing, Advised solenocyte collection; Toshihide Yamashita, Supervision, Funding acquisition, Writing—review and editing, Supervised the project

## Author ORCIDs

Rieko Muramatsu (iD) http://orcid.org/0000-0001-5342-7823

## Ethics

Human subjects: The research protocol was approved by the Human Use Review Committees of Toneyama National Hospital (TNH-2018031). Informed consent was obtained from all subjects.

## Decision letter and Author response

Decision letter https://doi.org/10.7554/eLife.41869.017
Author response https://doi.org/10.7554/eLife.41869.018

# Additional files

## Supplementary files

• Transparent reporting form
DOI: https://doi.org/10.7554/eLife.41869.013

## Data availability

All the representative data has been deposited to Dryad (10.5061/dryad.nj51t60).

The following dataset was generated:

| Author(s) | Year | Dataset title | Dataset URL | Database and Identifier |
|---|---|---|---|---|
| Machika Hamaguchi | 2019 | Data from: Circulating transforming growth factor-beta1 facilitates remyelination in the adult central nervous system | https://dx.doi.org/10.5061/dryad.nj51t60 | Dryad Digital Repository, 10.5061/dryad.nj51t60 |

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
