## [Decision Letter]

Thank you for sending your article entitled "Circulating transforming growth factor-beta1 facilitates remyelination in the adult central nervous system" for peer review at *eLife*. Your article is being evaluated by 3 peer reviewers, and the evaluation is being overseen by a guest Reviewing Editor and Didier Stainier as the Senior Editor.

Particular emphasis should be placed on experiments that provide direct evidence for a specific role of circulating transforming growth factor-beta1 in remyelination.

*Reviewer #1:*

Summary:

In this article Hamaguchi et al. aim to investigate the potential role of circulating factors, specifically TGF-β1, in the maturation of oligodendrocyte lineage cells for the promotion of remyelination in the central nervous system. The authors demonstrate that serum stimulation facilitates the maturation of oligodendrocytes, and TGF-β1 signaling, in particular, is required for serum-induced maturation. The authors report that both circulating TGF-β1 and TGF-βRI in oligodendrocyte lineage cells are required for normal spontaneous remyelination following LPC-induced demyelination. Additionally, TGF-β1 administration, by both cerebroventricular and intravenous injection, was demonstrated to promote remyelination in two complementary models of demyelination. The authors also report that human oligodendrocytes express TGF-βRI and stimulation of human oligodendrocyte cultures with recombinant TGF-β1 promotes the expression of myelin-associated genes. Hamaguchi et al. conclude that the administration of systemic TGF-β1 may hold therapeutic potential for treating demyelinating diseases by entering the CNS in conditions of increased blood brain barrier permeabilization. While the authors provide strong support for a role of TGF-β1 signaling in remyelination, their current evidence does not directly show that the therapeutic effect of circulating TGF-β1 requires entry into the CNS for the promotion of oligodendrocyte maturation.

Essential revisions:

1) This study builds upon previous evidence for the therapeutic potential of TGF-β1 in demyelinating diseases, but it does not fully resolve the long-standing debate about the therapeutic mechanism of TGF-β1 in demyelinating diseases as it claims. The authors provide strong evidence for TGF-β1 signaling promoting oligodendrocyte maturation, but the authors do not account for the influence of TGF-β1 on regulatory T cells in recovery from demyelination. To determine the therapeutic mechanism of circulating TGF-β1, authors should provide a direct link between the therapeutic effect of systemic administration of TGF-β1 and oligodendrocyte expressed TGF-βRI. To establish the therapeutic mechanisms of circulating TGF-β1, authors likely need to systemically administer TGF-β1 to mice with conditional deletion of TGF-βRI in oligodendrocytes following LPC-induced demyelination. Provided systemic administration of TGF-β1 has no therapeutic effect in mice lacking TGF-βRI in oligodendrocytes, authors may conclude that CNS entry of circulating TGF-β1 is required to promote remyelination.

2) A weakness of the present study is the use of anti-CD41-induced platelet depletion as the sole method for reducing circulating TGF-β1. The authors determine that depletion of platelets reduces circulating TGF-β1, but they do not account for altering other platelet-dependent effects on remyelination. An additional tool to reduce circulating TGF-β1 is required prior to concluding that circulating TGF-β1 is critical in remyelination.

*Reviewer #2:*

In the manuscript "Circulating transforming growth factor-β 1 facilitates remyelination in the adult central nervous system", Hamaguchi et al. describe the role of TGFbeta 1 on promoting oligodendrocyte differentiation and remyelination. The role of TGFbeta1 in these processes have previously been established, and the authors present data that indeed confirms the role of TGF-beta1 on oligodendrocyte maturation and remyelination. The authors suggest that circulating TGF-beta1 is important in remyelination, which would be a very interesting observation. Unfortunately, I don't consider that the data on the current manuscript is conclusive and the authors would need to provide much further substantiation for this hypothesis.

Essential revisions:

- The authors embark in a pharmacological screen, but the screen is barely mentioned in the results or methods. The authors need to describe the screen in much more detail. Also, it is not clear what is the rationale for focusing on TGF-beta1 rather than other pathways that appear to have a much more robust effect. Finally, the supplement table 1 is very poorly described, it is not clear what which column refers to, the authors need to explain properly this table.

- The authors deplete platelets using anti-CD41 antibodies and observe a reduction of MBP staining 14 days after LPC injection:

- Depletion of the platelets will most likely not only affect the transport of TGF-beta1, but also of other factors as VEGF, among others. As such, the authors cannot ascribe the effects only to TGF-beta1. A regain of function experiment, such as intracerebroventricular infusion of TGF-beta1 in a platelet depletion paradigm and upon PLC injection, would be able to validate the authors' hypothesis.

- The authors should validate the reduction of the levels of TGF-beta1 protein at the lesion, as in Figure 2C

- When referring to the decrease in MBP staining in the lesions, the authors often mention "demyelination area". One of the advantages of the LPC experimental setup is that one can differentiate demyelination from remyelination, as they are temporally separated in this paradigm. Since the authors are evaluating 14 days after LPC, they are most likely assessing remyelination. If they want to assess demyelination, they should assess earlier time points. The authors should revise their conclusions throughout the text and I would advise adding schemes of the experimental setup for data collection after LPC, so it is clear what outcome they are assessing.

- The inducible KO of TGF-beta1 receptor approach in Figure 4B-D reinforces the role of TGF-β 1 in remyelination, but does not address the role of circulating TGF-beta1, which would be the major novelty of the paper.

- The effects on systemic TGF-beta1 in EAE are interesting, but are modest, it is not clear whether the changes in MBP is due to effects on demyelination or remyelination, which is particularly relevant given that it has been shown that TGF-beta1 has an effect on immune cells in the context of EAE (Lee et al., 2015, among others).The authors should do a more thorough analysis on the putative effects of TGF-beta1 on the immune system in their paradigm, in order to be able to rule out that these are not the main mediators of the effects observed.

*Reviewer #3:*

Oligodendrocyte maturation is necessary for functional regeneration in the CNS. The authors show that TGF-beta1 is present in higher levels in the peripheral environment and promotes oligodendrocyte maturation. They show decreasing circulating TGF-β1 level reduces remyelination in the spinal cord in a toxin-induced demyelination model. TGF-β1 administration promotes remyelination and restored neurological function in a multiple sclerosis animal model. In addition, they show TGF- beta1 stimulates maturation of human oligodendrocyte in vitro.

The role of TGF-beta1 in oligodendrocyte differentiation and maturation has been well studied. The protective effects of TGF-beta1 has also been reported. The novel findings of this manuscript are that the circulating TGF-beta1 promotes oligodendrocyte maturation and facilitates remyelination. The experiments are overall well designed.

An important concern about the paper is how specific are the findings to oligodendrocytes and whether the behavioral deficits observed can be attributed only to oligodendrocyte maturation. I have the following comments:

Figure 1. To show TGF-beta1 enhances oligodendrocyte maturation directly, they authors need to show that oligodendrocytes express the receptors of TGF-β and that TGF-β1 activates the signaling pathway in oligodendrocytes (for example increase of pSamd2,3 or downstream target gene expression).

Figure 2. It is interesting that the spleen has highest concentration of TGF-beta1, is it because the spleen produces more than other organs or because TGF-beta1 accumulates in the spleen?

In addition, some important data rely on quantitative analyses of immunostaining. It would improve the quality of the manuscript if those data would also be supported by an independent assay.

---

## [Author Response]

Reviewer #1:

[…] Essential revisions:1) This study builds upon previous evidence for the therapeutic potential of TGF-β1 in demyelinating diseases, but it does not fully resolve the long-standing debate about the therapeutic mechanism of TGF-β1 in demyelinating diseases as it claims. The authors provide strong evidence for TGF-β1 signaling promoting oligodendrocyte maturation, but the authors do not account for the influence of TGF-β1 on regulatory T cells in recovery from demyelination. To determine the therapeutic mechanism of circulating TGF-β1, authors should provide a direct link between the therapeutic effect of systemic administration of TGF-β1 and oligodendrocyte expressed TGF-βRI. To establish the therapeutic mechanisms of circulating TGF-β1, authors likely need to systemically administer TGF-β1 to mice with conditional deletion of TGF-βRI in oligodendrocytes following LPC-induced demyelination. Provided systemic administration of TGF-β1 has no therapeutic effect in mice lacking TGF-βRI in oligodendrocytes, authors may conclude that CNS entry of circulating TGF-β1 is required to promote remyelination.

According to the reviewer’s suggestion, we injected LPC into the spinal cords of oligodendrocyte (OL)-specific TGF-βRI knockout mice (PLP-Cre::TGF-βRI flox) and treated the mice with recombinant mouse TGF-β1. Immunohistochemical analysis revealed that TGF-β1 treatment did not change the MBP-negative area in the spinal cord 2 weeks after LPC injection (Figure 4E,F), indicating that CNS entry of circulating TGF-β1 is required to promote remyelination. We added these results to subsection “TGF-βRI in oligodendrocytes is involved in spontaneous remyelination”.

2) A weakness of the present study is the use of anti-CD41-induced platelet depletion as the sole method for reducing circulating TGF-β1. The authors determine that depletion of platelets reduces circulating TGF-β1, but they do not account for altering other platelet-dependent effects on remyelination. An additional tool to reduce circulating TGF-β1 is required prior to concluding that circulating TGF-β1 is critical in remyelination.

We treated wild-type mice with TGF-β neutralizing antibodies and injected LPC into the mice spinal cords. We evaluated the demyelinated area by measuring the MBPnegative area in spinal cord sections. Immunohistochemical analysis revealed that mice treated with TGF-β neutralizing antibodies had a larger MBP-negative area than controls (i.e., mice subjected to control IgG treatment, Figure 3F,G), consistent with the results of the platelet depletion experiments (Figure 3B,C). We added the results in subsection “Circulating TGFβ-1 contributes to remyelination in the CNS”.

Reviewer #2:

[…] Essential revisions:- The authors embark in a pharmacological screen, but the screen is barely mentioned in the results or methods. The authors need to describe the screen in much more detail. Also, it is not clear what is the rationale for focusing on TGF-beta1 rather than other pathways that appear to have a much more robust effect. Finally, the supplement table 1 is very poorly described, it is not clear what which column refers to, the authors need to explain properly this table.

Thank you for this feedback. We screened 160 drugs and found 107 drugs (blue area Panel 1 of Author response image 1) that did not decrease the abundance of Olig2positive cells (0.5 times lower than control) in the culture after each treatment. We next excluded the drugs that decreased MBP expression (0.5 times lower than control) in the cells by drug treatment or serum treatment (without drug), and identified 17 drugs (please see orange area Panel 2 of Author response image 1) that downregulated MBP area (2 times lower than control) relative to cells subjected to serum treatment. In this experiment, we searched for a circulating molecule that drives oligodendrocyte maturation directly; therefore, we focused on the TGF-β receptor type I inhibitor (TGF-βRI), the only drug we tested that specifically targets a receptor-type protein. We added this description to the Results section. We also revised Figure 1—source data 1 and the figure legend of source data 1.

- The authors deplete platelets using anti-CD41 antibodies and observe a reduction of MBP staining 14 days after LPC injection:- Depletion of the platelets will most likely not only affect the transport of TGF-beta1, but also of other factors as VEGF, among others. As such, the authors cannot ascribe the effects only to TGF-beta1. A regain of function experiment, such as intracerebroventricular infusion of TGF-beta1 in a platelet depletion paradigm and upon PLC injection, would be able to validate the authors' hypothesis.

Thank you for this suggestion. We believe that the reviewer is asking us whether other platelet-derived factors are involved in remyelination. To answer this question, we must evaluate whether platelet-specific TGF-β1 depletion decreases remyelination efficiency after LPC injection. However, platelet-specific depletion is technically difficult because plateletspecific Cre mice are not available. Hence, as an alternative, we used neutralizing TGF-β1 antibodies. We found that TGF-β1 neutralization prevented remyelination after LPC injection (Figure 3F,G). In addition, we compared the inhibitory ratio of remyelination between TGF-β1 neutralization and platelet depletion, and found that they were comparable (*Pint* = 0.6126, single regression analysis), indicating that TGF-β1 makes a major contribution to remyelination mediated by circulating factors. We added this description in the Results section (Subsection “Circulating TGFβ-1 contributes to remyelination in the CNS”).

- The authors should validate the reduction of the levels of TGF-beta1 protein at the lesion, as in Figure 2C.

We evaluated the intensity of the TGF-β1 signal around the LPC lesion in spinal cords from mice treated with or without CD41 antibodies. Immunohistochemical analysis revealed that mice treated with CD41 antibodies had lower levels of TGF-β1 than controls (Figure 3—figure supplement 1B,C), indicating that depletion of platelets decreased the level of TGF-β1 at the lesion. We added this description to the Results section (subsection “Circulating TGFβ-1 contributes to remyelination in the CNS”).

- When referring to the decrease in MBP staining in the lesions, the authors often mention "demyelination area". One of the advantages of the LPC experimental setup is that one can differentiate demyelination from remyelination, as they are temporally separated in this paradigm. Since the authors are evaluating 14 days after LPC, they are most likely assessing remyelination. If they want to assess demyelination, they should assess earlier time points. The authors should revise their conclusions throughout the text and I would advice adding schemes of the experimental setup for data collection after LPC, so it is clear what outcome they are assessing.

Thank you for the suggestion. As reviewer #2 might have expected, we did not focus on demyelination in this study. We emphasized our focus on remyelination in the Results section (subsection “Circulating TGFβ-1 contributes to remyelination in the CNS”) and revised “demyelinated area” to “MBP negative area” in pertinent passages.

- The inducible KO of TGF-beta1 receptor approach in Figure 4B-D reinforces the role of TGF-β 1 in remyelination, but does not address the role of circulating TGF-beta1, which would be the major novelty of the paper.

To investigate the effect of circulating TGF-β1 on remyelination, we used TGF-β neutralizing antibodies and investigated whether TGF-β neutralization would prevent remyelination after LPC injection. Immunohistochemical analysis revealed that the MBPnegative area was higher in mice treated with TGF-β neutralizing antibodies than in controls (Figure 3F,G), consistent with results obtained using oligodendrocyte-specific TGF-βRI knockout mice (Figure 4B,C). This consistency in the data supports our conclusion that circulating TGF-β1 promotes spontaneous remyelination in this context.

- The effects on systemic TGF-beta1 in EAE are interesting, but are modest, it is not clear whether the changes in MBP is due to effects on demyelination or remyelination, which is particularly relevant given that it has been shown that TGF-beta1 has an effect on immune cells in the context of EAE (Lee et al., 2015, among others).The authors should do a more thorough analysis on the putative effects of TGF-beta1 on the immune system in their paradigm, in order to be able to rule out that these are not the main mediators of the effects observed.

According to the reviewer’s suggestion, we investigated whether TGF-β1 regulates inflammatory responses, which are related to disease progression of EAE (Muramatsu et al., 2012). To this end, we collected lymphocytes and splenocytes from EAE mice and re-stimulated the cells with MOG_35-55_peptides in vitro. Evaluation of cytokine expression and cell proliferation activity revealed that TGF-β1 did not change the inflammatory response in our experimental contexts (Figure 5—figure supplement 1C,D). These data confirm our observation that TGF-β1-mediated remyelination does not depend on regulation of immunological response. We added this explanation to the Results section.

Reviewer #3:

[…] An important concern about the paper is how specific are the findings to oligodendrocytes and whether the behavioral deficits observed can be attributed only to oligodendrocyte maturation. I have the following comments:Figure 1. To show TGF-beta1 enhances oligodendrocyte maturation directly, they authors need to show that oligodendrocytes express the receptors of TGF-β and that TGF-β1 activates the signaling pathway in oligodendrocytes (for example increase of pSamd2,3 or downstream target gene expression).

Thank you for this suggestion. We investigated the expression of TGF-β1 receptor in the mouse oligodendrocyte culture. Immunocytochemical analysis revealed that APClabeled oligodendrocytes express TGF-β1 receptor. We provided a representative image in Figure 1J and revised the Results section accordingly. To determine whether TGFβ1 activates signaling pathway in oligodendrocytes, we stimulated mouse oligodendrocytes with recombinant TGF-β1 and evaluated phosphorylation of the Smad2 protein. Western blot analysis revealed an increase in Smad2 phosphorylation 30 mins after TGF-β1 treatment of oligodendrocytes (Figure 1—figure supplement 1F,G). We added this information to the Results section.

Figure 2. It is interesting that the spleen has highest concentration of TGF-beta1, is it because the spleen produces more than other organs or because TGF-beta1 accumulates in the spleen?

To determine why the spleen had the highest concentration of TGF-β1, we depleted platelets in the mice by injecting CD41 antibodies, and then measuring the TGF-β1 level in the spleen. TGF-β1 levels in spleen were lower in mice subjected to platelet depletion than in controls (Figure 3—figure supplement 1A), suggesting that a high concentration of TGF-β1 in the spleen depends on platelet-mediated TGF-β1 accumulation in the spleen. We added this result on subsection “Circulating TGFβ-1 contributes to remyelination in the CNS”.

In addition, some important data rely on quantitative analyses of immunostaining. It would improve the quality of the manuscript if those data would also be supported by an independent assay.

According to the reviewer’s suggestion, we performed electron microscopic analysis to determine whether our observation by immunostaining encompass structural remyelination. For this purpose, we injected platelet-depleted mice with LPC and found that the thickness of myelin sheaths was significantly lower in mice treated with anti-CD41 mAb than in controls (Figure 3D,E), indicating that our findings were also consistent with structural remyelination. We added this result on subsection “Circulating TGFβ-1 contributes to remyelination in the CNS”. We appreciate the reviewer’s suggestions because the additional experiments we performed in response to these comments have strengthened the core argument of this manuscript.